# In Vivo-Matured Oocyte Resists Post-Ovulatory Aging through the Hub Genes *DDX18* and *DNAJC7* in Pigs

**DOI:** 10.3390/antiox13070867

**Published:** 2024-07-19

**Authors:** Cheng-Lin Zhan, Dongjie Zhou, Ming-Hong Sun, Wen-Jie Jiang, Song-Hee Lee, Xiao-Han Li, Qin-Yue Lu, Ji-Dam Kim, Gyu-Hyun Lee, Jae-Min Sim, Hak-Jae Chung, Eun-Seok Cho, Soo-Jin Sa, Xiang-Shun Cui

**Affiliations:** 1Department of Animal Science, Chungbuk National University, Cheongju 28644, Republic of Korea; zcl@chungbuk.ac.kr (C.-L.Z.); dongjiez@uio.no (D.Z.); sunminghong19940216@swu.edu.cn (M.-H.S.); jwj@neau.edu.cn (W.-J.J.); songhee@chungbuk.ac.kr (S.-H.L.); xhli@chungbuk.ac.kr (X.-H.L.); surpassway@chungbuk.ac.kr (Q.-Y.L.); jidom0331@chungbuk.ac.kr (J.-D.K.); coacoacoa4171@chungbuk.ac.kr (G.-H.L.); tlawoals99@cbnu.ac.kr (J.-M.S.); 2The Center for Reproductive Control, TNT Research Co., Ltd., Jiphyeonjungang 3-gil 13, Sejong-si 30141, Republic of Korea; hjchung@tntresearch.co.kr; 3Swine Science Division, National Institute of Animal Science, RDA, Cheonan-si 31000, Republic of Korea; segi0486@korea.kr; 4Planning and Coordination Division, National Institute of Animal Science, Iseo-myeon, Wanju-gun 55365, Republic of Korea; soojinsa@korea.kr

**Keywords:** post-ovulatory aging, oocyte quality, mitochondrial function, post-transcriptional regulation

## Abstract

Assisted reproduction technology (ART) procedures are often impacted by post-ovulatory aging (POA), which can lead to reduced fertilization rates and impaired embryo development. This study used RNA sequencing analysis and experimental validation to study the similarities and differences between in vivo- and vitro-matured porcine oocytes before and after POA. Differentially expressed genes (DEGs) between fresh in vivo-matured oocyte (F_vivo) and aged in vivo-matured oocyte (A_vivo) and DEGs between fresh in vitro-matured oocyte (F_vitro) and aged in vitro-matured oocyte (A_vitro) were intersected to explore the co-effects of POA. It was found that “organelles”, especially “mitochondria”, were significantly enriched Gene Ontology (GO) terms. The expression of genes related to the “electron transport chain” and “cell redox homeostasis” pathways related to mitochondrial function significantly showed low expression patterns in both A_vivo and A_vitro groups. Weighted correlation network analysis was carried out to explore gene expression modules specific to A_vivo. Trait–module association analysis showed that the red modules were most associated with in vivo aging. There are 959 genes in the red module, mainly enriched in “RNA binding”, “mRNA metabolic process”, etc., as well as in GO terms, and “spliceosome” and “nucleotide excision repair” pathways. *DNAJC7*, *IK*, and *DDX18* were at the hub of the gene regulatory network. Subsequently, the functions of *DDX18* and *DNAJC7* were verified by knocking down their expression at the germinal vesicle (GV) and Metaphase II (MII) stages, respectively. Knockdown at the GV stage caused cell cycle disorders and increase the rate of abnormal spindle. Knockdown at the MII stage resulted in the inefficiency of the antioxidant melatonin, increasing the level of intracellular oxidative stress, and in mitochondrial dysfunction. In summary, POA affects the organelle function of oocytes. A_vivo oocytes have some unique gene expression patterns. These genes may be potential anti-aging targets. This study provides a better understanding of the detailed mechanism of POA and potential strategies for improving the success rates of assisted reproductive technologies in pigs and other mammalian species.

## 1. Introduction

Oocyte quality and maturation not only affect the embryo and fertilization success but also have long-term impacts on fetal growth and development. Notably, mature oocytes arrested at metaphase II (MII) can be successfully fertilized within a restricted time window after ovulation. When fertilization fails to occur within this optimal time frame, MII oocytes undergo a process of degradation known as “post-ovulatory aging” (POA) [1]. As time passes, the cytoplasm undergoes aging and degenerates, which is an inevitable process [1]. In mice, if oocytes are not fertilized within six hours after ovulation, their quality deteriorates rapidly, and they are subsequently lost after 12 h [2]. For human oocytes, the optimal time for fertilization is within a time window of 4–12 h after ovulation [3]. However, women do not exhibit natural visible signs of ovulation, and fertilization likely occurs hours later after ovulation, which will cause the oocyte to age before combining with the spermatozoa.

POA has many negative effects on oocytes, including hardening of the zona pellucida [4], partial exocytosis of cortical granules [4,5], abnormalities in the cytoskeleton, and condensation of the chromosome [6]. POA can also cause a decline in maturation-promoting factor and mitogen-activated protein kinase (MAPK) levels [7], mitochondrial dysfunction, and apoptosis [8,9,10], and perturbation of Ca^2+^ homeostasis; oxidative damage to lipids, proteins, and DNA components of the cell [11]; and epigenetic changes [12]. Additionally, POA can lower the rate of fertilization and embryo quality while increasing the risk of offspring abnormalities. Therefore, the timely fertilization of oocytes is crucial to ensure embryonic development. Although oocytes fertilized via intracytoplasmic sperm injection (ICSI) can develop into viable embryos, the procedure itself takes 6–12 h and can induce POA [13]. Aging-induced defects such as oxidative stress, mitochondrial dysfunction, and chromosomal abnormalities have been detected in oocytes during assisted reproductive technology (ART) procedures, which can impact oocyte quality, leading to lower fertilization rates and aberrant embryo development, and even unhealthy offspring [14,15,16]. Therefore, developing strategies to improve the anti-aging ability of oocyte outcomes remains imperative.

Currently, there is no research available that compares the anti-aging ability of in vivo- and vitro-matured porcine oocytes. Compared to other species, porcine oocytes are very similar to human oocytes in many ways. For instance, they share a similar closed oocyte volume of 120–125 μm in diameter, and a similar average time for oocyte maturation, this being 40–44 h for pigs and 40 h for humans [17]. Both species also require a similar core transcriptional network to maintain pluripotency [18] and have a similar developmental stage of embryonic genome activation at the four- to eight-cell stage [17,19]. Additionally, both porcine and human oocytes contain a large amount of endogenous lipids [20,21]. For these reasons, porcine oocytes are an excellent model for human reproductive research and clinical-assisted reproductive technology applications. Therefore, we collected in vivo- and vitro-matured and aged porcine oocytes for RNA sequencing to explore the effects of POA on porcine oocytes and the key pathways and genes that provide in vivo-matured oocytes with stronger anti-aging ability.

In this study, we used transcriptomic analysis and experimental validation to study the similarities and differences between in vivo- and vitro-matured porcine oocytes during POA. By answering this question, we will gain a better understanding of the specific mechanisms of POA and provide potential strategies for improving the success rates of assisted reproductive technologies in pigs and other mammalian species.

## 2. Methods

### 2.1. Collection of Porcine Cumulus–Oocyte Complexes (COCs) and In Vitro Maturation (IVM)

Porcine ovaries were collected from a local slaughterhouse (Farm Story Dodarm B&F, Umsung, Chungbuk, Republic of Korea) and transported to the laboratory in a pre-warmed NaCl solution containing 75 mg/mL penicillin G and 50 mg/mL streptomycin sulfate. Porcine follicles approximately 3–6 mm in diameter were aspirated using a 10 mL disposable syringe. COCs with more than two layers of compact cumulus cells (CCs) were selected and washed three times with an IVM medium (TCM-199 [11150–059; Gibco, Grand Island, NY, USA] supplemented with 100 mg/L sodium pyruvate, 10 ng/mL EGF, 10% (*v*/*v*) porcine follicular fluid, 10 IU/mL LH, and 10 IU/mL FSH). Finally, 50–100 COCs per well were cultured in four-well dishes covered with mineral oil for 44–48 h until maturation to the MII phase at 38.5 °C with 5% CO_2_.

### 2.2. In Vitro Fresh and Aged Oocyte Collection

In vitro fresh and aged oocytes were collected as described previously [22,23]. CCs were removed with 1 mg/mL hyaluronidase by pipetting approximately 40 times after IVM. MII stage oocytes were selected for further studies. To analyze oocyte POA, the selected MII stage oocytes were continuously cultured in an IVM medium covered with mineral oil for an additional 48 h.

### 2.3. In Vivo—Fresh and Aged—Oocyte Collection

In vivo oocytes were collected as described previously [24]. All crossbred gilts (Duroc) were housed at the National Institute of Animal Science (NIAS, Cheonan, Republic of Korea). The gilts were synchronized using Altrenogest ReguMate or Virbagest for 15 d. Briefly, all animals were detected as being in estrus by back-pressure testing and were kept in separate pens without insemination for mature oocyte collection. On day 2 post-estrus, the gilts were stunned by electro-anesthesia, slaughtered by bleeding in a commercial slaughterhouse, and their reproductive organs were obtained immediately after slaughter. The mature oocytes were then flushed from the oviduct with 10 mL of pre-warmed PBS, after which the oocytes were carefully selected in a Petri dish under a stereomicroscope. For POA, the selected MII stage oocytes were continuously cultured in an IVM medium for an additional 48 h at 38.5 °C in a humidified atmosphere (5% CO_2_). All samples were removed from the zona pellucidae using an acid Tyrode’s solution and collected in a tube, each containing five oocytes. The samples were then stored at −80 °C until library preparation.

### 2.4. RNA Extraction

mRNA was isolated using the Dynabeads mRNA Direct Kit (61012; Thermo Fisher Scientific, Waltham, MA, USA) from 10 MII stage oocytes in each sample, three replicate samples per group. The in vivo oocytes were derived from 4 donor sows. the obtained oocytes were mixed and randomly and evenly grouped. cDNA was synthesized using the First Strand Synthesis Kit (cat# 6210; LeGene, San Diego, CA, USA) following the manufacturer’s instructions. 

### 2.5. Quantitative Reverse Transcription PCR (qRT-PCR)

qRT-PCR was performed using a WizPure qPCR Master (W1731-8; Wizbio Solutions, Seongnam, Republic of Korea) according to the manufacturer’s instructions, on a Quant Studio™ 6 Flex Real-Time PCR System (Applied Biosystems, Waltham, MA, USA). The PCR conditions were as follows: initial denaturation at 95 °C for 10 min, followed by 40 cycles of amplification at 95 °C for 15 s, 60 °C for 20 s, and 72 °C for 15 s, and a final extension at 95 °C for 15 s. Relative gene expression was calculated using the 2^−∆∆CT^ method. A standard curve was constructed using a 10-fold dilution method to verify that the amplification efficiency was 95.72% (Appendix A). The primers used in this study are listed in Table 1.

### 2.6. Construction of cDNA Libraries and RNA Sequencing

RNA sequencing was performed with five MII stage oocytes in each sample, three replicate samples per group, and the in vivo oocytes were derived from 4 donor sows. The obtained oocytes were mixed and randomly and evenly grouped. Total RNA was extracted using a TRIzol reagent kit (Invitrogen, Carlsbad, CA, USA) according to the manufacturer’s protocol. RNA quality was assessed using an Agilent 2100 Bioanalyzer (Agilent Technologies, Palo Alto, CA, USA) and verified by RNase-free agarose gel electrophoresis. After total RNA extraction, the eukaryotic mRNA was enriched with Oligo(dT) beads. The enriched mRNA was fragmented into short fragments using a fragmentation buffer and reverse transcribed into cDNA using the NEBNext Ultra RNA Library Prep Kit for Illumina (NEB #7530; New England Biolabs, Ipswich, MA, USA). The purified double-stranded cDNA fragments were end-repaired, and a base was added and ligated to Illumina sequencing adapters. The ligation reaction was purified using AMPure XP Beads (1.0X) and PCR was performed to amplify the ligation products. The resulting cDNA library sequencing was performed on the Illumina HiseqTM 2500/4000 by Gene Denovo Biotechnology Co., Ltd. (Guangzhou, China) with a sequencing strategy of Paired-end 150 bp (PE150). RNA sequencing data analysis procedures and parameters are shown in the Appendix A.

### 2.7. Preparation of DDX18 and DNAJC7 Double-Stranded RNA (dsRNA)

*DDX18* and *DNAJC7* cDNA fragments were amplified from total cDNA of porcine cumulus cells using primers containing the T7 promoter sequence (Table 1). In vitro transcription was performed using the MEGAscript T7 Kit (AM1333; Thermo Fisher Scientific, Waltham, MA, USA) to synthesize dsRNA according to the manufacturer’s instructions. After 10 h of in vitro transcription, the dsRNA mixture was treated with DNase I for 15 min to degrade the DNA template and then purified using Riboclear™ Plus (313–150; GeneAll Biotechnology, Seoul, Republic of Korea). The purified dsRNA was dissolved in RNase-free water and stored at −80 °C until use.

### 2.8. dsRNA Injection

COCs were microinjected after 2 h of IVM. MII stage oocytes with the first polar body were selected after 44 h of IVM, and COCs with extended cumulus cells were pipetted 20–30 times with 1 mg/mL hyaluronidase to remove the cumulus cells. COCs and oocytes were microinjected with 5–10 pL of nucleus-free water (NC group) or 1200 ng/μL dsRNA (KD group) using an Eppendorf Femto-Jet (Eppendorf, Hamburg, Germany) under a Nikon Diaphot Eclipse TE300 inverted microscope (Nikon, Tokyo, Japan).

### 2.9. Measurement of Mitochondrial Membrane Potential (ΔΨm), Reactive Oxygen Species (ROS), and Glutathione (GSH)

To determine ΔΨm, the MII oocytes were treated with 2.5 µM 5,5′,6,6′-tetrachloro-1,1′,3,3′-tetraemyl-imidacarbocyanine iodide (JC-1) (M34152; Thermo Fisher Scientific) in the PBS/PVA for 30 min at 38.5 °C. The ratio of the intensity of red fluorescence of the activated mitochondria (J-aggregates) to that of green fluorescence of less-activated mitochondria (J-monomers) was calculated as the membrane potential. To determine total ROS levels, MII oocytes were incubated for 30 min at 38.5 °C in PBS/PVA containing 10 μM 2′,7′-dichlorodihydrofluorescein diacetate (H2DCF-DA, Cat # D399, Molecular Probes, Eugene, OR, USA). To determine GSH levels, MII oocytes were cultured for 30 min at 38.5 °C in PBS/PVA containing 10 μM 4-chloromethyl-6,8-difluoro-7-hydroxycoumarin dye (CellTracker™ Blue CMF2HC, Thermo Fisher Scientific, Waltham, MA, USA). After incubation, the MII oocytes were washed thrice with PBS/PVA. Fluorescence signals were detected using a digital camera (DP72; Olympus, Tokyo, Japan) connected to a fluorescence microscope (IX70; Olympus). The fluorescence signal was analyzed using ImageJ v.l.52i software (National Institutes of Health, Bethesda, MD, USA).

### 2.10. Immunofluorescence Staining

At room temperature (RT), 10–15 oocytes were collected and fixed in 3.7% formaldehyde for 1 h. After washing thrice in phosphate-buffered saline (PBS) containing 0.1% PVA (PBS/PVA) for 5 min, the embryos were permeabilized with 0.1% Triton X-100 for 30 min at RT. The embryos were then washed thrice and blocked with PBS/PVA containing 3% BSA for 1 h. Subsequently, the oocytes were incubated with α-Tubulin antibody (Santa cruz, Dallas, TX, USA, sc-5286) or cytochrome c antibody (Abcam, ab110325, Cambridge, UK) overnight at 4 °C. After washing with PBS/PVA, the oocytes were cultured with Alexa Fluor 488™ donkey anti-mouse IgG (1:200) and Alexa Fluor 568™ donkey anti-mouse IgG (1:200) at 37 °C for 1 h. After washing thrice, the oocytes were mounted onto glass slides using an antifade mounting medium containing DAPI (Vector Laboratories, Inc., Newark, CA, USA), and examined under a confocal microscope (LSM-880, Zeiss, Jena, Germany). Images were obtained using Zen software (version 3.10; Zeiss).

### 2.11. Statistical Analysis

Each experiment was performed three times, each sample in triplicate. Data were analyzed using one-way analysis of variance (ANOVA) or the Student’s *t*-test. All percentage data were subjected to arcsine transformation before statistical analysis and are presented as mean ± SEM. Differences were considered statistically significant at *p* < 0.05. All calculations were performed using the GraphPad Prism 10 software (GraphPad, San Diego, CA, USA).

## 3. Results

### 3.1. Global Transcriptome Analysis of Fresh and Aged In Vivo and Vitro-Matured Porcine Oocytes

As shown in Figure 1, GV stage oocytes were collected from pig ovaries and cultured for 44 and 92 h to obtain fresh in vitro-matured oocytes (F_vitro) and aged in vitro-matured oocytes (A_vitro). Fresh in vivo-matured oocytes (F_vivo) and aged in vivo-matured oocytes (A_vivo) were obtained by in vitro culturing of MII stage oocytes, collected from porcine fallopian tubes, for 0 and 48 h, respectively.

To investigate the differentiations in gene expression in response to POA, RNA-seq analysis was performed on the four groups of oocytes. For each sample, 62–89 million row reads were generated (Appendix A). Low-quality data and rRNA reads were filtered to obtain clean data. Subsequently, ≥76.77% of the clean data were mapped to the reference genome (Appendix A). Furthermore, ≥83.28% of total mapped reads were compared to the exon region (Appendix A). The expression value for each gene was calculated as the expected number of fragments per kilobase of transcript sequence per million base pairs sequenced (FPKM). A total of 25,479 expressed genes in porcine oocytes were identified based on a cut-off of FPKM > 0.0 (Appendix A). A violin plot of log10-transformed FPKM values for each sample revealed that these samples had consistent overall ranges and distributions of the FPKM values (Appendix A). Then, we detected the relative expression levels of randomly selected genes *BTF3*, *CEP95*, *CMC1*, *DBI*, and *NOUFB5* to verify the accuracy of transcriptome sequencing by qPCR, and we found that the expression trends were consistent with the sequencing results (Appendix A). The results indicated that the RNA sequencing data obtained in this study were reliable, reproducible, and of high quality.

Principal component analysis showed that oocytes from the four groups were in different sets (PC1: 43.5%, PC2: 27.1%). The four groups showed significant differentiation and good reproducibility (Figure 2A). The Spearman correlation heatmap showed good intra-group reproducibility and the four groups were clearly distinguished from each other (Figure 2B). The heat map (Figure 2C) shows overall gene expression patterns across four groups. These results indicate that different cultural environments had a profound influence on the quality and anti-aging abilities of porcine oocytes. As shown in Figure 2D, 3989 genes (1482 up and 2507 down) were differentially expressed between the F_vivo and A_vivo, 5880 genes (3006 up and 2874 down) were differentially expressed between the F_vivo and F_vitro, 1130 genes (928 up and 202 down) were differentially expressed between the A_vivo and A_vitro, and 1535 genes (642 up and 893 down) were differentially expressed between the F_vitro and A_vitro (Appendix A).

### 3.2. POA Interferes with the Mitochondrial Function in Both In Vivo- and Vitro-Matured Porcine Oocytes

To exclude the great interference of the culture environment, DEGs between F_vivo and A_vivo and DEGs between F_vitro and A_vitro were intersected to explore the co-effects of POA (Figure 3A). To better understand the roles the DEGs played during the process of POA, the DEGs were enriched into Gene Ontology (GO) terminology. The top 15 enriched GO terms were shown in Figure 3B. The top enriched terms were mainly related to Intracellular, Organelle, and Mitochondrion. In addition, we found that the expression of electron transport chain-related genes *POLG2*, *FDX1*, *GLRX2*, *COX8C*, etc., and cell redox homeostasis-related genes *TXNDC11*, *PRDX3*, and *GLRX2* decreased in both A_vivo and A_vitro groups (Figure 3C,D). In addition, the results of Kyoto Encyclopedia of Genes and Genomes (KEGG) pathway enrichment analysis showed that the oxidative phosphorylation pathway was significantly enriched (Figure 4A). The expression of genes related to oxidative phosphorylation pathway *COX6B1*, *NDUFB5*, *UQCRQ*, etc., were downregulated in both A_vivo and A_vitro groups (Figure 4D). These results indicate that POA can cause damage to oocyte mitochondrial function and affect intracellular energy supply and redox status.

### 3.3. POA Affects Protein Export and Oocyte Meiosis in Both In Vivo- and Vitro-Matured Porcine Oocytes

The top 15 enriched KEGG pathways are shown in Figure 3B. Ribosome, protein export, spliceosome, and thermogenesis were the most enriched pathways. In addition, we also found that the oocyte meiosis pathway which is closely related to oocyte maturation was enriched. The heat map shows protein export and meiosis-related gene expression patterns. Genes related to protein export including *SRP54*, *SPCS2*, *SEC11C*, etc., were significantly downregulated in both A_vivo and A_vitro. The expression levels of *CDK1*, *ANAPC10*, and *SPDYA* were lower in both A_vivo and A_vitro groups. The expression levels of *PKMYT1* and *PPP2R1A* in the aged group were lower than the F_vitro but higher than F_vivo group. These results indicate that POA can lead to impaired oocyte protein export function and affect meiosis progression.

### 3.4. DNAJC7, DDX18 and IK Were the Hub Genes in the Gene Regulation Network in Aged In Vivo Porcine Oocytes

The sample clustering dendrograms were constructed based on the genes for which FPKM > 2 (total of 11,804 genes). The applicable power value for this test was nine. Then, gene modules were detected based on the TOM matrix. A total of 22 modules were detected in the analysis (Figure 5A), coded as brown, black, red, pink, green, grey60 module, etc., including 3136, 1705, 959, 773, 716, and 655 genes, respectively Appendix A.

Next, we used module feature values and specific trait data to perform correlation analysis to find the modules most relevant to traits and phenotypes. The most significant module based on the correlation of the modules with the traits was the red module (Figure 5B) (Appendix A). A total of 959 target genes are included within the red module, and the expression level of genes within the red module is shown in Figure 5C. Most genes expressed higher in the A_vivo group. 

The gene significance (GS) and module membership (MM) values of genes were used to analyze the association between each trait and the module, and the module with high correlation played an important biological role in the trait. The intramodular connectivity (K.in) and GS values of genes within the module were used to analyze the association between the module and the gene and the trait (Appendix A). The result shows that the red module has a high correlation and connectivity with traits. Therefore, the red module was selected for further analysis.

We further analyzed the 959 candidate hub genes in the red module by GO/KEGG enrichment analysis. Ten significantly enriched GO terms were shown in Appendix A. RNA binding, mRNA metabolic process, RNA splicing, DNA repair, etc., were enriched. Ten significantly enriched KEGG pathways were shown in Appendix A. Spliceosome, nucleotide excision repair, DNA replication, RNA polymerase, etc., were enriched. These enrichment results are highly correlated, suggesting that oocytes matured in vivo may resist POA through RNA binding, modification, splicing, DNA repair, etc., which are related to post-transcriptional gene expression.

The gene regulatory relationship network diagram was used to obtain the core genes at the hub position in the regulatory relationship network within the red module (Figure 5D). DnaJ heat shock protein family (Hsp40) member C7 (*DNAJC7*), IK cytokine *(IK*), DEAD-Box Helicase 18 (*DDX18*) and transcription factors zinc finger proteins 836 (*ZNF836*), and *ZNF667* were at the hub position. The expression patterns of these five genes are shown in Figure 5E. All five genes have higher expression levels in the A_vivo than in the A_vitro group.

### 3.5. Knockdown of DNAJC7 and DDX18 at GV Stage Resulted in Cell Cycle Disorders and Abnormal Spindle Morphology in Porcine Oocytes

To explore the functions of the selected hub genes, the expression of *DDX18* and *DNAJC7* was knocked down by injecting the dsRNA into the GV stage COCs (Figure 6A). After 44 h, MII oocytes were collected for qPCR and it was determined that the mRNA levels of *DDX18* and *DNAJC7* decreased by 82.52% and 71.64%, respectively (Figure 6B). Subsequently, all oocytes were collected after 44 h of IVM, and we performed α-Tubulin staining to determine their meiotic stages. Representative images of each stage are shown in Figure 6C. Knockdown of *DDX18* and *DNAJC7* resulted in more oocytes staying in the GVBD and MI stages (Figure 6D). Normal spindle morphology at the MII stage is critical for the resumption of meiosis after fertilization and subsequent embryonic development [25,26]. Higher rates of abnormal spindles were found in *DDX18* and *DNAJC7* knockdown oocytes (Figure 6E,F). These results confirm that *DDX18* and *DNAJC7* were involved in regulating the cell cycle and spindle formation during oocyte maturation.

### 3.6. Knockdown of DNAJC7 and DDX18 at MII Stage Resulted in Oxidative Stress and Mitochondrial Dysfunction in Porcine Oocytes

To further determine the role of *DDX18* and *DNAJC7* during the POA process, the expression of *DDX18* and *DNAJC7* was knocked down by injecting the dsRNA into the MII stage oocytes. At the same time, melatonin, an antioxidant that has been widely proven to reduce oxidative stress in porcine oocytes [27,28], was added to the extended 48-h IVM (Figure 7A). During the POA process, MII stage oocytes will exhibit abnormal phenotypes such as cytoplasmic fragmentation, second polar body exclusion, cleavage, and unclear cell membrane boundaries (Figure 7B). The addition of melatonin significantly increased the proportion of normal oocytes after POA, but the knockdown of *DDX18* and *DNAJC7* rendered melatonin ineffective (Figure 7C). Next, the changes in intracellular ROS, GSH, mitochondrial membrane potential, and cytochrome c were detected to measure the level of oxidative stress and mitochondrial function (Figure 7D). Intracellular ROS levels increased significantly after POA, and the supplementation of melatonin was able significantly to reduce ROS, while the knockdown of *DDX18* and *DNAJC7* reduced the ROS scavenging effect of melatonin (Figure 7E). Similar results were also shown in the changes in GSH content. The intracellular GSH level was significantly reduced after POA, and melatonin increased the GSH content. Knockdown of DDX18 and DNAJC7 reduced GSH content (Figure 7F). In terms of mitochondrial function, mitochondrial membrane potential was reduced by POA, and melatonin increased it. This rescue effect was also offset by the knockdown of DDX18 and DNAJC7 (Figure 7G). Cytochrome c was an essential component of the electron transport chain [29]. Its expression was significantly reduced after POA. The addition of melatonin increased its expression in senescent oocytes but was ineffective in senescent oocytes with the knockdown of *DDX18* and *DNAJC7*. These results indicate that *DDX18* and *DNAJC7* were critical in oocyte resistance to oxidative stress and mitochondrial function damage caused by POA.

## 4. Discussion

The exact mechanisms underlying POA are poorly understood. Previous studies have compared the differences in gene expression profiles in GV or MII stage oocytes from young and old individuals [30,31], which were limited by the differences in in vivo and in vitro culture environments that are hard to ignore. Herein, we used RNA sequencing to determine the gene expression profiles of oocytes in the following four groups: fresh in vivo, fresh in vitro, aged in vivo, and aged in vitro. The similar and different effects of POA on in vivo- and vitro-matured oocytes were explored. 

Most DEGs appeared between fresh in vivo and in vitro oocytes, with a total of 5880. Different cultural environments had a profound influence on the quality and anti-aging abilities of porcine oocytes. To exclude the influence of in vitro conditions, DEGs between F_vivo and A_vivo and DEGs between F_vitro and A_vitro were intersected to explore the co-effects of POA.

These results showed that mitochondrial dysfunction, including the electron transport chain, cell redox homeostasis, and oxidative phosphorylation, contributed to the aging of oocytes. During oocyte maturation, continuous transcription and translation require a significant amount of ATP, making the availability of an appropriate number of functional mitochondria crucial [32]. It has been suggested that higher ATP content in human and mouse MII oocytes is associated with better embryo potential for development and implantation [33]. The normal function of mitochondria ensures oocyte quality and embryo developmental potential, while mitochondrial dysfunction and energy metabolism disturbance are related to oocyte aging [34]. This highlights the importance of mitochondrial function in aging. 

POA can also cause abnormalities in some oocyte-specific cellular processes, such as germinal vesicle breakdown (GVBD) and meiosis. Abnormal distribution of mitochondria as well as mitochondrial dysfunction, results in severely impaired GVBD of mouse oocytes [35]. This conclusion was also confirmed by our results, as KEGG enrichment analysis showed that the meiosis pathway was enriched, and the expression of related genes was reduced in aging oocytes. These results were consistent with our previous study that found POA can cause an increase in oxidative stress levels and a decrease in MII rate, blastocyst rate, ATP content, mitochondria DNA copy number, and mitochondrial biogenesis-related proteins in porcine oocytes [22,23].

Another enriched pathway in both in vivo- and vitro-matured oocytes was protein export. This pathway contains a form of DEGs named signal recognition particle (SRP), which is a ribonucleoprotein particle crucial for co-translational targeting of secretory and membrane proteins to the endoplasmic reticulum [36]. The eukaryotic SPC is composed of five subunits including two isoforms of catalytic subunits SEC11A and SEC11C and three regulatory subunits including signal peptidase complex subunit 1 (SPCS1), SPCS, and SPCS3 [37,38]. Among the SPC subunits, SEC11 and SPCS3 are essential for signal peptidase activity and cell survival [39,40]. A previous study has shown that nuclear protein export is a common hallmark of pathological and physiological aging in the Hutchinson–Gilford syndrome cellular phenotype of normal fibroblasts [41].

WGCNA was performed to explore the unique gene expression patterns of in vivo-matured oocytes during POA. A total of 22 co-expression network modules were determined and successfully mined the specific module related to A_vivo. WGCNA divides modules into soft thresholds, which reflect the effectiveness of biological networks more effectively than hard thresholds [42]. Functional enrichment analysis using GO/KEGG identified RNA binding and spliceosome as key pathways against POA in vivo.

RNA-binding proteins (RBPs) achieve their biological function essentially by post-transcriptional gene regulation [43]. In eukaryotic cells, following transcript synthesis, RBPs dictate extensive pre-mRNA processing by interacting with the target RNA and partner proteins [44]. They are responsible for adding modifications to the transcript that affect stability and translation efficiency, including 5’-end capping and 3’-end polyadenylations. RBPs play a crucial role in the transport of transcripts from the nucleus to the cytoplasm, where protein synthesis occurs [45,46].

Decades of work on aging have shed light on the fundamental role played within this context by a class of proteins termed RBPs [47,48]. Loss of intracellular RBP AU-rich-element factor-1 will alter post-transcriptional regulation of targets particularly relevant for the protection of genomic integrity and gene regulation, thus concurring with responses related to oxidative stress and accelerated aging [49]. RBP PUM1 overexpression protected MSCs against H_2_O_2_-induced cellular senescence by suppressing TLR4-mediated NF-κB activity [50]. 

Spliceosomes induced by pharmacological and genetic inhibition of spliceosome genes have been reported to trigger cell senescence, suggesting a key role of spliceosome genes as a gatekeeper [51]. Alternative splicing generates diverse transcripts by removing introns and splicing exons. Pre-mRNA splicing is fundamental for gene expression and regulation. Similar results were observed in transcriptome sequencing data from human mature defective oocytes, with spliceosomes being the most abundant pathway [52]. 

The hub genes *DNAJC7*, *DDX18*, and *IK* were screened out by constructing a gene regulatory relationship network. These hub genes are involved to a certain extent in maintaining genome stability, DNA damage repair, and other life processes. *DNAJC7* participates in the *p53/MDM2* negative feedback regulatory pathway, thereby enhancing the stability and activity of tumor suppressor *p53* which promotes apoptosis to eliminate cells with seriously damaged DNA to maintain genomic integrity [53]. *DDX18* depletion leads to γH2AX accumulation and genome instability [54]. *IK* is a splicing factor that promotes spliceosome activation and contributes to pre-mRNA splicing [55]. All three genes had higher expression levels in the A_vivo than the A_vitro group.

These results demonstrate the importance of these hub genes on cellular senescence. In vivo-matured oocytes have higher expression levels than in vitro-matured oocytes. We predict that in vivo-matured oocytes can resist POA through these hub genes.

Thus, dsRNA was firstly used to knockdown the expression of *DDX18* and *DNAJC7* at the GV stage. More oocytes were arrested in the GVBD and MI stages after knocking down. At the same time, there was a significantly higher abnormal rate of the spindle morphology of oocytes at the MIII stage. This indicates that *DDX18* and *DNAJC7* have functions in maintaining normal cell cycle and spindle morphology. Then, the expression of *DDX18* and *DNAJC7* was knocked down at the MII stage. An extended 48 h of IVM was undertaken to explore their role in the POA process. The addition of some antioxidants such as Epigallocatechin-3-gallate [22], melatonin [56], and Ubiquinol-10 [23] had good preventive effects on POA and was able to improve the fertilization ability of in vitro oocytes and the developmental ability of mammal embryos. Melatonin addition was used to simulate in vivo-matured oocytes by reducing damage caused by POA. The results showed that the addition of melatonin effectively increased the proportion of normal oocytes, reduced intracellular oxidative stress levels, and improved mitochondrial function. However, these rescue functions were counteracted by the *DDX18* and *DNAJC7* knockdowns. These results confirm that deletion of *DDX18* and *DNAJC7* leads to intracellular oxidative stress and mitochondrial dysfunction, reducing the effectiveness of traditional antioxidants such as melatonin. Therefore, it is important to develop novel antioxidants based on these potential gene targets.

Our findings indicate that POA affects the quality of porcine oocytes, likely due to its impact on mitochondrial function and protein export. We observed variations in the expression patterns of in vivo- and vitro-matured oocytes during POA, especially in pathways related to post-transcriptional regulation, such as RNA-binding and spliceosome pathways, in which *DDX18* and *DNAJC7* were the hub genes. (Figure 8). By utilizing bioinformatics data and analysis methodologies and functional verification experiments, pathways and key genes *DDX18* and *DNAJC7* have been identified that may be involved in the POA process. The identification of these target pathways and genes is potentially helpful for the development of new anti-POA antioxidants or methods to protect oocytes affected by aging during the ART process.

## 5. Conclusions

Our research has revealed the considerable influence of POA on the quality of the porcine oocyte. The impairment of mitochondrial function and protein export mechanisms appears to be at the core of these adverse outcomes. These mechanisms are crucial for maintaining the cellular integrity and developmental potential of oocytes.

The RNA-binding and spliceosome pathways have been identified as the most significantly enriched pathways when comparing oocytes that matured in vivo with those matured in vitro. This difference in pathway enrichment highlights the distinct nature of embryos matured in vivo versus in vitro when encountering POA and suggests potential mechanisms that could be targeted to reduce the negative effects of POA.

The identification of hub genes, including *DNAJC7*, *DDX18*, and *IK*, suggests that these genes play crucial roles in the regulation of post-transcriptional modification and the maintenance of mitochondrial function, thus representing key points of intervention for improving oocyte quality affected by POA. Loss of *DDX18* and *DNAJC7* results in cell cycle disorders during oocyte maturation and reduced effectiveness of traditional antioxidants such as melatonin during POA. This underlines the need to develop novel antioxidants based on these potential mechanisms. 

Present research adds to the increasing understanding of the molecular processes involved in POA. It provides a basis for developing strategies to mitigate the adverse effects of POA on porcine oocytes. By targeting these molecular mechanisms, it may be feasible to enhance the quality and developmental capacity of oocytes, ultimately improving the success rates of assisted reproductive technologies in pigs and potentially other mammalian species.

## Figures and Tables

**Figure 1 antioxidants-13-00867-f001:**
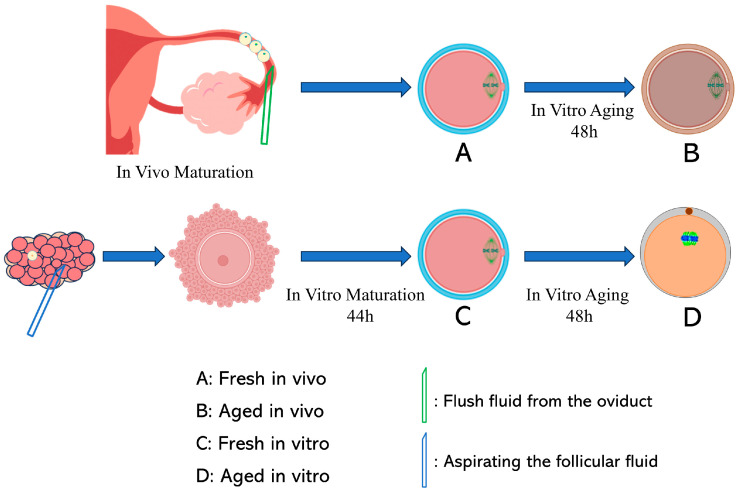
Experimental flowchart. In brief, GV stage oocytes were collected from porcine ovaries and cultured for 44 and 92 h to obtain two groups of oocytes, fresh in vitro and aged in vitro, respectively. Fresh in vivo and aged in vivo oocytes were obtained by in vitro culture of MII stage oocytes collected from porcine fallopian tubes for 0 and 48 h, respectively.

**Figure 2 antioxidants-13-00867-f002:**
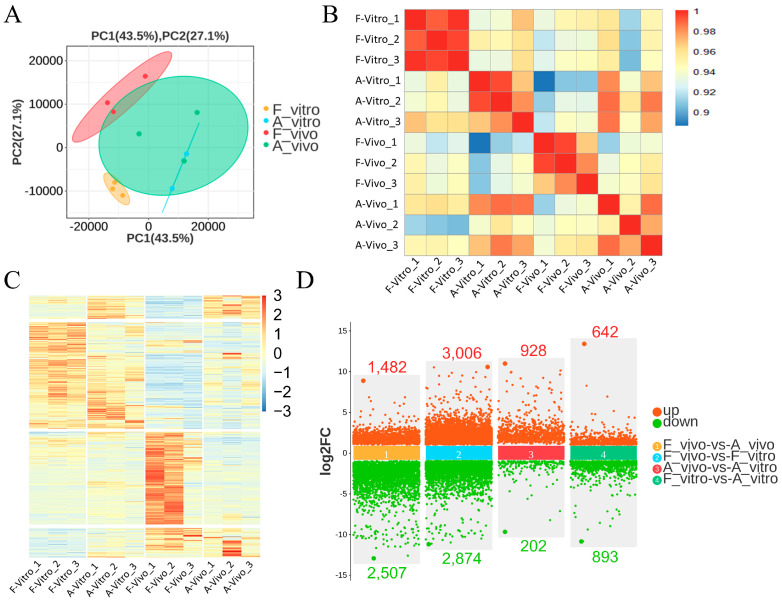
Global transcriptome analysis of fresh and aged in vivo and vitro-matured porcine oocytes. (**A**) Principal component analysis of four groups, all groups showed good grouping and repeatability. PC1 = 43.5%, PC2 = 27.1%. (**B**) Spearman’s correlation heatmap shows the reproducibility between repetitive samples and the difference between groups. The color gradient indicates the magnitude of the correlation coefficient. (**C**) Heatmap of mRNA expression profiles in fresh and aging oocytes, showing changes in a subset of genes in response to POA. The color key (from blue to red) of the Z-score value indicates low to high expression levels. (**D**) Scatterplot of differentially expressed genes between each two groups. The criteria for differentially expressed genes were FDR < 0.05, and fold change > 2.

**Figure 3 antioxidants-13-00867-f003:**
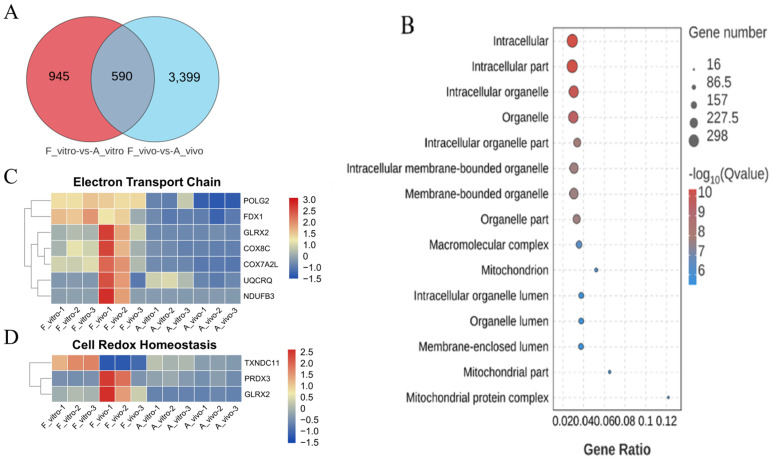
GO enrichment analysis of common DEGs caused by POA in both in vivo and vitro-matured oocytes. (**A**) Differential gene Venn diagram. The intersection of DEGs between F_vivo and A_vivo and DEGs between F_vitro and A_vitro were defined as common DEGs. (**B**) GO enrichment analysis on common DEGs showing the top 15 enriched GO items. (**C**) The heat map for gene expression levels associated with electron transport chain in enriched GO terms. (**D**) The heat map for gene expression levels associated with cell redox homeostasis in enriched GO terms.

**Figure 4 antioxidants-13-00867-f004:**
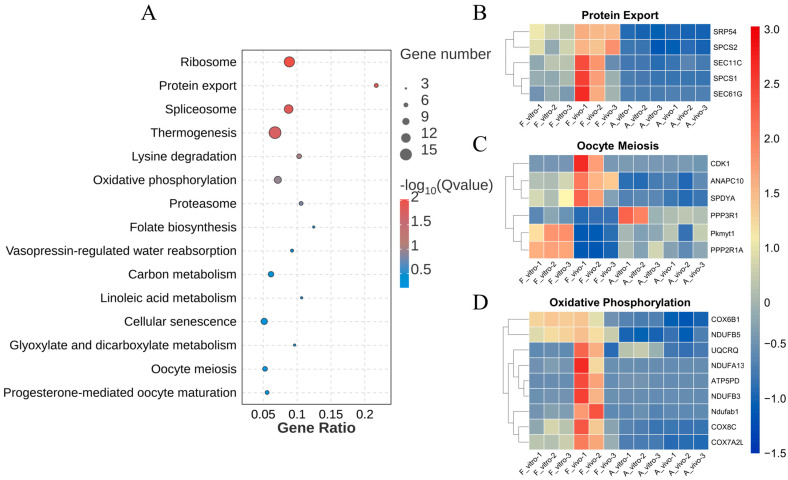
KEGG enrichment analysis of common DEGs caused by POA in both in vivo- and vitro-matured oocytes. (**A**) KEGG enrichment analysis on common DEGs showing the top 15 enriched KEGG pathways. (**B**) The heat map for gene expression levels associated with protein export in enriched KEGG pathway. (**C**) The heat map for gene expression levels associated with oocyte meiosis in enriched KEGG pathway. (**D**) The heat map for gene expression levels associated with oxidative phosphorylation in enriched KEGG pathway.

**Figure 5 antioxidants-13-00867-f005:**
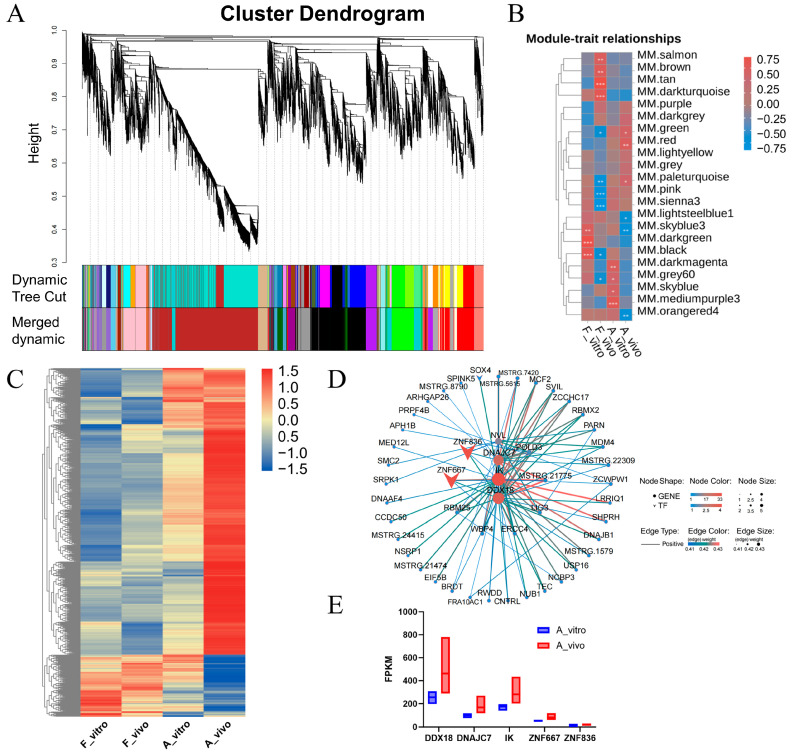
Screening of gene modules and hub genes specifically expressed in in vivo matured aging oocytes. (**A**) Module hierarchical clustering tree. Gene modules are divided according to the clustering relationship between genes. Genes with similar expression patterns will be classified into the same module. The branches of the cluster tree are cut and distinguished to generate different modules. Each color represents a module, and gray indicates genes that cannot be assigned to any module. After the preliminary module division, the preliminary divided “Dynamic Tree Cut” is obtained. Modules with similar expression patterns are then merged based on the similarity of module feature values to obtain the final dividend “Merged dynamic”. (**B**) Module–trait relationships. Each row presents a module eigengene, each column presents a trait. Each cell contains the corresponding correlation and *p*-value (* means *p*-value < 0.05, ** means *p*-value < 0.01, *** means *p*-value < 0.001). The table is color-coded by correlation according to the color legend. (**C**) The heat map for gene expression levels of genes in the red module among the four groups. (**D**) Construction of gene regulatory network. Each node in the figure is a gene, and each line represents the regulatory relationship between the nodes. The darker and larger the node color is, the higher the abundance and the stronger the connectivity. The weight value defines the color and thickness of the line. The darker the color and the thicker the line, the stronger the regulatory relationship between genes. (**E**) The boxplot shows the expression of hub genes in the A_vivo and A_vitro groups.

**Figure 6 antioxidants-13-00867-f006:**
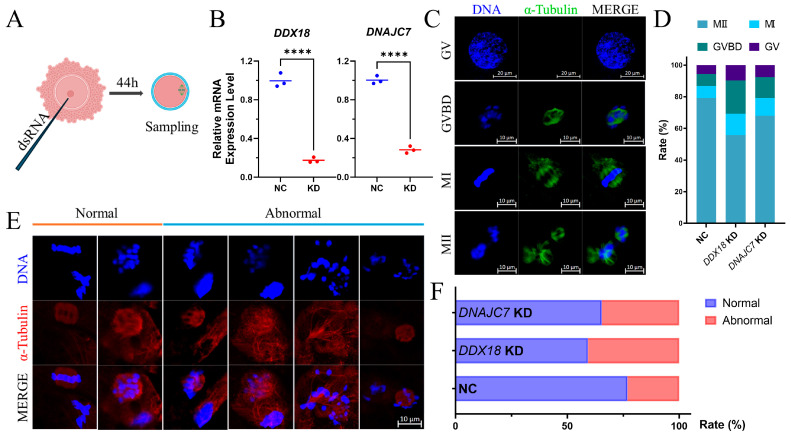
Effects of *DDX18* and *DNAJC7* knockdown on oocyte maturation. (**A**) Knockdown of *DDX18* and *DNAJC7* expression by injecting dsRNA into GV stage COCs. (**B**) Relative mRNA expression levels of *DDX18* and *DNAJC7* at MII stage oocytes. (**C**) Representative images of α-Tubulin morphology in GV, GVBD, MI, and MII stages. Bar = 10 μL and 20 μL in the GV line. (**D**) The proportion of oocytes at each stage after *DDX18* and *DNAJC7* knockdown. (**E**) Representative images of normal and abnormal α-Tubulin morphology in MII stage oocytes. Bar = 10 μL. (**F**) The proportion of normal and abnormal α-Tubulin morphology in NC, *DDX18* KD, and *DNAJC7* KD group MII stage oocytes. **** means *p*-value < 0.0001.

**Figure 7 antioxidants-13-00867-f007:**
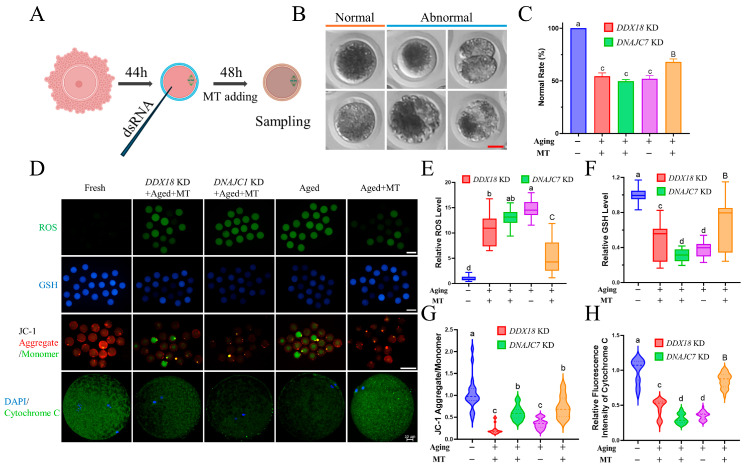
Effects of *DDX18* and *DNAJC7* knockdown on oocyte maturation. (**A**) The expression of *DDX18* and *DNAJC7* was knocked down at the MII stage, and then melatonin was added during an extended 48-h culture. (**B**) Representative images of normal and abnormal oocyte morphology after POA. Bar = 50 μL. (**C**) The proportion of normal oocyte morphology after *DDX18* and *DNAJC7* knockdown and melatonin addition. (**D**) Representative images of ROS, GSH, JC-1, and cytochrome c fluorescent staining. Bar = 200 μL and 20 μL in DAPI/cytochrome c line. (**E**) Changes in ROS levels after *DDX18* and *DNAJC7* knockdown and melatonin addition. (**F**) Changes in GSH contents after *DDX18* and *DNAJC7* knockdown and melatonin addition. (**G**) Changes in mitochondrial membrane potential after *DDX18* and *DNAJC7* knockdown and melatonin addition. (**H**) Changes in cytochrome c protein expression levels after *DDX18* and *DNAJC7* knockdown and melatonin addition. Differences among means were assessed using a multiple comparison test. Data without the same letters means significantly different. Lowercase letters indicate *p* < 0.05, and uppercase letters indicate *p* < 0.01.

**Figure 8 antioxidants-13-00867-f008:**
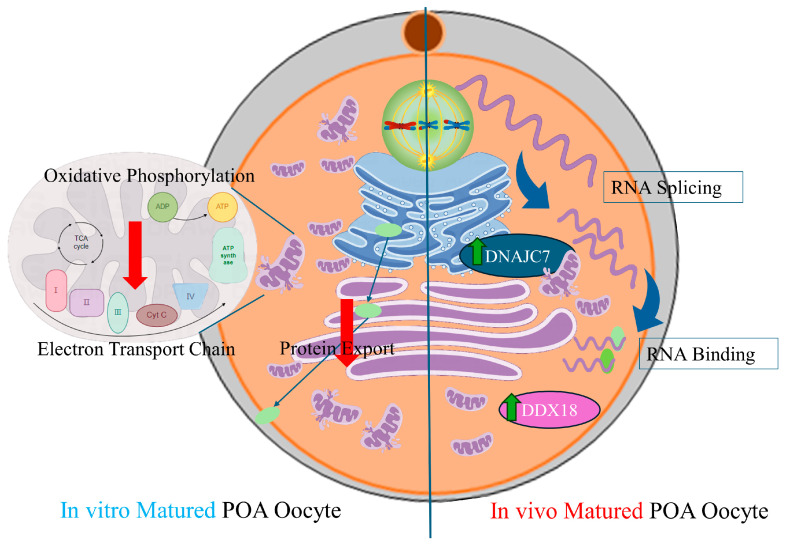
Schematic representation depicting the same and different effects of POA on in vivo- and vitro-matured porcine oocytes. POA adversely affects the quality of porcine oocytes, both in vivo and in vitro. This is most likely due to the impairment (Red arrows) of mitochondrial function and protein export. RNA-binding and spliceosome pathways were the most differentially enriched pathways, and *DDX18* and *DNAJC7* were the differentially expressed hub genes and higher (Green arrows) expressed in oocytes that matured in vivo than those that matured in vitro.

**Table 1 antioxidants-13-00867-t001:** Information of primers used in this study.

Gene	Full Name	Primer Sequence (5′~3′)	Gene ID
*BTF3*	Basic transcription factor 3	F: GTGTGTGCGCCTTATCTCAG R: GTTTGGCGAGTTTCTCCTGG	100514824
*CEP95*	Centrosomal protein 95	F: AGAGGGCAGGAGAGAGGTTA R: ACATCCTCCTCTTCACAGCC	100624364
*CMC1*	C-X9-C motif containing 1	F: CGCAGAACAGCATCTCAGAC R: TCCAGAGTCCTTGCAGCATT	102164646
*DBI*	Diazepam binding inhibitor	F: ACAGCCACTACAAACAAGCG R: ACGCTTTCATGGCATCTTCC	397212
*NDUFB5*	NADH: ubiquinone oxidoreductase subunit B5	F: GCTTTGCCCTCAGTCAACAT R: CATGGCTACTATGGGCGAGA	100523751
*DDX18*	DEAD-box helicase 18	F: GCCAACACGCAGACAGACCAT R: CCTGCTCAAGACCATCCACTGT	100153560
		* F: TGCGACAGTGGATGGTCTTGAG * R: TGCTTCCTGGGCTGACTTATGA	
*DNAJC7*	DnaJ heat shock protein family (Hsp40) member C7	F: CCTGCCTGCCATCGCTTCAA R: AGCATCTGCGTTGGTGGAATCC	100524895
		* F: GGATTCCACCAACGCAGATGCT * R: TCCTTCTGAACTTCCGCACTGG	
*18s*	18S ribosomal RNA	F: CGCGGTTCTATTTTGTTGGT R: AGTCGGCATCGTTTATGGTC	100861538

* indicates that the primer is used for dsRNA production, otherwise it is used for qPCR.

## Data Availability

The original contributions presented in the study are included in the article, and further inquiries can be directed to the corresponding author.

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
