# Peer review of "In Vivo-Matured Oocyte Resists Post-Ovulatory Aging through the Hub Genes DDX18 and DNAJC7 in Pigs"

_antioxidants, 2024, doi:10.3390/antiox13070867_

Round 1

Reviewer 1 Report (Previous Reviewer 1)

The manuscript was very much improved. In the current version, methods are clear, and connect to results, discussion and conclusion.

This is an important study, especially now a days, when pig industry is seeking to implement embryo production 

I have no other comments, all my issues were solved.

I just suggest a carefull reading, since in some places in the text English might be improved and nomemclature for the genes are not in Italics

Author Response

Reviewer 2 Report (Previous Reviewer 2)

It is difficult to evaluate the current manuscript without major revesion on the previous comments.

They have been fixed.

Author Response

Reviewer 3 Report (New Reviewer)

In the present work, the authors want to study the pathways involved in oocyte aging used for invitro fertilization. Certainly is very important to now the key step involved in the permanency of oocyte healthy for fertilization and the impact that they can have for future experimental procedures. It is a carefully done work but I have some small doubt to include in the manuscript

1.- How the primers were selected for the analysis of the samples?  Did you have a suspect of the putative pathways affected?

2.- How the authors implement the methods?. Did they previously used a previously implemented methodology?. If so, please include references of the working group in which they originally explained the methods used

3.- The authors use two different methods to isolated RNA, one with Trizol and the other with microbeads. Why the use two methods. Is one of them superior that the other? Because of the importance to have a very well purified sample for this analysis, could you explain please?

4.- It is a good methodological and descriptive work to obtain the possible pathways affected during aging of postovulatory oocytes. In this regard, the authors need to analyses if the enriched pathways are expected as a normal aging step of cells or if there are some pathway that is distinctive for the aging oocyte.  It needs a discussion to balance the new information to respond to the hypothesis.

5.- There are good experimental evidences to suggest many pathway affected in the aging of postovulatory oocytes. The discussion could be organized as a suggestion on the putative pathway main affected during the aging of the oocytes. The work could be an expanding series of work to confirm the descriptive work of the paper and to expand the knowledge on an extremely complicated work that could have impact to the IVF procedures

The authors use two different methods to isolated RNA, one with Trizol and the other with microbeads. Why the use two methods. Is one of them superior that the other? Because of the importance to have a very well purified sample for this analysis, could you explain please?

The authors need to explain how the primers were selected for the analysis of the samples. 

Author Response

Reviewer 4 Report (New Reviewer)

Since the problem of oocyte aging is critical and affects IVF outcomes in human IVC practice, this research is very actual. The model animal is correctly selected, because porcine embryos are similar to human ones in many properties. The authors chose modern experimental methods, like transcriptomic analysis for RNA sequencing and sophisticated analytical approaches to investigate oocyte aging process both in vivo and in vitro. Such experimental design, to compare anti-aging ability of in vivo- or in vitro- matured oocytes, has not been reported yet. The authors proved the importance of proper gene expression, protein export and mitochondrial functionality in the oocyte ageing process. This manuscript, therefore, brings novel findings in the task of mammalian oocyte aging.

Generally, the manuscript is well-written, however a minor English correction is needed, like:

 “help us understand” – lines 35, 85;

“dysfunction of mitochondrial function” – line 449;

several sentences are beginning form the word “And…”, which is not correct;

and other places in the text.

I have only doubt, whether given manuscript is topically related to the Antioxidant journal area, because no antioxidant effects are mentioned in the study.

Author Response

This manuscript is a resubmission of an earlier submission. The following is a list of the peer review reports and author responses from that submission.

Round 1

Reviewer 1 Report

It is a relevant paper, however, as I described in the boxes above, a lot of improvement at the methods and Results section has to be made, so the reader can access the information more easilly

Details are already at the boxes above

Reviewer 2 Report

It has been well known that mitochondria function is impaired after aging. Also most data have been provided just by bioinformatics based on RNA seq by NGS without any functional analysis or even confirmation using different methods such as qPCR for DNAJC7 etc. More functional and physiological analysis and confirmation of the results should be necessary to convince the significance of authors’ findings. 

1.    P4 Lines145-, authors should provide all the methods of bioinformatics including WGCNA.

2.    P9 Line 270, “Indicates” should be “indicates”.

3.    P1 Line 30, “the red modules” is supposed to be one of modules. I could not understand why the authors refer to the “red” module in the abstract. Any specific meaning is implied in “red”.    
